# Asynchronous Stochastic Subgradient Methods for General Nonsmooth Nonconvex Optimization

## Abstract

Asynchronous distributed methods are a popular way to reduce the communication and synchronization costs of large-scale optimization. Yet, for all their success, little is known about their convergence guarantees in the challenging case of general non-smooth, non-convex objectives, beyond cases where closed-form proximal operator solutions are available. This is all the more surprising since these objectives are the ones appearing in the training of deep neural networks.

In this paper, we introduce the first convergence analysis covering asynchronous methods in the case of general non-smooth, non-convex objectives. Our analysis applies to stochastic sub-gradient descent methods both with and without block variable partitioning, and both with and without momentum. It is phrased in the context of a general probabilistic model of asynchronous scheduling accurately adapted to modern hardware properties. We validate our analysis experimentally in the context of training deep neural network architectures. We show their overall successful asymptotic convergence as well as exploring how momentum, synchronization, and partitioning all affect performance.

## 1 Introduction

Training parameters arising in Deep Neural Net architectures is a difficult problem in several ways (Goodfellow et al., 2016). First, with multiple layers and nonlinear activation functions such as sigmoid and softmax functions, the ultimate optimization problem is nonconvex. Second, with ReLU activation functions and max-pooling in convolutional structures, the problem is nonsmooth, i.e., it is not differentiable everywhere, although typically the set of non-differentiable points is a set of measure zero in the space of the parameters. Finally, in many applications it is unreasonable to load the whole sample size in memory to evaluate the objective function or (sub)gradient, thus samples must be taken, necessitating analysis in a probabilistic framework.

The analysis of parallel optimization algorithms using shared memory architectures, motivated by applications in machine learning, was ushered in by the seminal work of Recht et al. (2011) (although precursors exist, see the references therein). Further work refined this analysis, e.g. (Liu & Wright, 2015) and expanded it to nonconvex problems, e.g. (Lian et al., 2015). However, in all of these results, a very simplistic model of asynchronous computation is presented to analyze the problem. Notably, it is assumed that every block of the parameter, among the set of blocks of iterates being optimized, has a fixed, equal probability of being chosen at every iteration, with a certain vector of delays that determine how old each block is that is stored in the cache relative to the shared memory. As one can surmise, this implies complete symmetry with regards to cores reading and computing the different blocks. This does not correspond to asynchronous computation in practice. In particular, in the common Non-Uniform Memory Access (NUMA) setting, practical experience has shown that it can be effective for each core to control a set of blocks. Thus, the choice of blocks will depend on previous iterates, which core was last to update, creating probabilistic dependence between the delay vector and the choice of block. This exact model is formalized in Cannelli et al., which introduced a new probabilistic model of asynchronous parallel optimization and presented a coordinate-wise updating successive convex approximation algorithm.

In this paper, we are interested in studying parallel asynchronous stochastic subgradient descent for general nonconvex nonsmooth objectives, such as the ones arising in the training of deep neural network architectures. Currently, there is no work in the literature specifically addressing this problem. The closest related work is given by Zhu et al. (2018) and Li & Li (2018), which consider asynchronous proximal gradient methods for solving problems of the form $f(x) + g(x)$, where $f$ is smooth and nonconvex, and $g(x)$ is nonsmooth, with an easily computable closed form prox expression. This restriction applies to the case of training a neural network which has no ReLUs or max pooling in the architecture itself, i.e., every activation is a smooth function, and there is an additional regularization term, such as an $\ell_1$. These papers derive expected rates of convergence. In the general case, where the activations themselves are nonsmooth—for instance in the presence of ReLUs—there is no such additive structure, and no proximal operator exists to handle away the non-smoothness and remove the necessity of computing and using subgradients explicitly in the optimization procedure.

This general problem of nonsmooth nonconvex optimization is already difficult (see, e.g., Bagirov et al. (2014)), and the introduction of stochastically uncertain iterate updates creates an additional challenge. Classically, the framework of stochastic approximation, with stochastic estimates of the subgradient approximating elements in a differential inclusion that defines a flow towards minimization of the objective function, is a standard, effective approach to analyzing algorithms for this class of problems. Some texts on the framework include Kushner & Yin (2003), which we shall reference extensively in the paper, and Borkar (2008). See also Ermol'ev & Norkin (1998) and Ruszczyński (1987) for some classical results in convergence of stochastic algorithms for nonconvex nonsmooth optimization. Interest in stochastic approximation has resurfaced recently sparked by the popularity of Deep Neural Network architectures. For instance, see the analysis of nonconvex nonsmooth stochastic optimization with an eye towards such models in Davis et al. (2018) and Majewski et al. (2018).

In this paper, we provide the first analysis for nonsmooth nonconvex stochastic subgradient methods in a parallel asynchronous setting, in the stochastic approximation framework. For this, we employ the state of the art model of parallel computation introduced in Cannelli et al., which we map onto the analysis framework of Kushner & Yin (2003). We prove show that the generic asynchronous stochastic subgradient methods considered are convergent, with probability 1, for nonconvex nonsmooth functions. This is the first result for this class of algorithms, and it combines the state of the art in these two areas, while extending the scope of the results therein. Furthermore, given the success of momentum methods (see, e.g., Zhang et al. (2017)), we consider the momentum variant of the classical subgradient method, again presenting the first convergence analysis for this class of algorithms.

We validate our analysis numerically by demonstrating the performance of asynchronous stochastic subgradient methods of different forms on the problem of ResNet deep network training. We shall consider variants of asynchronous updating with and without write locks and with and without block variable partitioning, showing the nuances in terms of convergence behavior as depending on these strategies and properties of the computational hardware.

## 2 PROBLEM FORMULATION

Consider the minimization problem

$$\min_x f(x), \tag{1}$$

where $f : \mathbb{R}^n \to \mathbb{R}$ is locally Lipschitz continuous (but could be nonconvex and nonsmooth) and furthermore, it is computationally infeasible to evaluate $f(x)$ or an element of the Clarke subdifferential $\partial f(x)$.

The problem (1) has many applications in machine learning, including the training of parameters in deep neural networks. In this setting, $f(x)$ is loss function evaluated on some model with $x$ as its parameters, and is dependant on input data $A \in \mathbb{R}^{n \times m}$ and target values $y \in \mathbb{R}^m$ of high dimension, i.e., $f(x) = f(x; (A, y))$, with $x$ a parameter to optimize with respect to the loss function. In cases of practical interest, $f$ is decomposable in finite-sum form,

$$f(x) = \frac{1}{M} \sum_{i=1}^{M} l(m(x; A_i); y_i)$$

where $l : \mathbb{R}^m \times \mathbb{R}^m \to \mathbb{R}$ represents the training loss and $\{(A_i, y_i)\}$ is a partition of $(A, y)$.

We are concerned with algorithms that solve (1) in a distributed fashion, i.e., using multiple processing cores. In particular, we are analyzing the following *inconsistent read* scenario: before computation begins, each core $c$ is allocated a block of variables $I^c$, for which it is responsible to update. At each iteration the core modifies a block of variables $i^k$, chosen randomly among $I^c$. Immediately after core $c$ completes its $k$-th iteration, it updates the shared memory. A lock is only placed on the shared memory when a core writes to it, thus the process of reading may result in computations of the function evaluated at variable values that never existed in memory, e.g., block 1 is read by core 1, then core 3 updates block 2, then core 1 reads block 2, and now block 1 is operating on a vector with the values in blocks 1 and 2 not simultaneously at their present local values at any point in time in shared memory. We shall index iterations to indicate when a core writes a new set of values for the variable into memory.

We let $\mathbf{d}^k = \{d_1^{k_c}, ..., d_n^{k_c}\}$ be the vector of delays for each component of the variable used to evaluate the subgradient estimate, thus the $j$-th component of $x$ that is used in the computation of the update at $k$ is actually not $x_j^{k_c}$ but $x_j^{d_j^{k_c}}$ .

In this paper, we are interested in applying stochastic approximation methods, of which the classic stochastic gradient descent forms a special case. Since $f$ in (1) is in general nonsmooth, we will exploit subgradient methods. Denote by $\xi^k$ the set of mini-batches used to compute an element of the subgradient $g((x_1^{d_1^{k_c}}, ..., x_n^{d_n^{k_c}}); \xi^{k_c})$. The set of minibatches $\xi^{k_c}$ is chosen uniformly at random from $(A, y)$, independently at each iteration. By the central limit theorem, the error is asymptotically Gaussian as the total size of the data as well as the size of the mini-batches increases.

**Asynchronous System Specification.** We consider a shared-memory system with $p$ processes concurrently and asynchronously performing computations on independent compute-cores. We interchangeably use the terms process and core. The memory allows concurrent-read-concurrent-write (CRCW)[1] access. The shared-memory system offers word-sized atomic `read` and `fetch-and-add (faa)` primitives. Processes use `faa` to update the components of the variable.

## 2.1 ALGORITHM DESCRIPTION

We now recall the stochastic subgradient algorithm under asynchronous updating in Algorithm 1, from the perspective of the individual cores. The update of the iterate performed by

$$u_{i^{k_c}} = m u_{i^{k_c}} + g_{i^{k_c}}^{k_c}; \ x_{i^{k_c}}^{k_c+1} = x_{i^{k_c}}^{k_c} - (1-m)\gamma^{k_c} u_{i^{k_c}}$$

where $m$ is the momentum constant, required to satisfy $0 < m < 1$

---

**Algorithm 1** Asynchronous Stochastic Subgradient Method for an Individual Core

**Input:** $x_0$, core $c$.
1: **while** Not converged **do**
2:    Sample $i$ from the variables $I_c$ corresponding to $c$. Sample $\xi$.
3:    Read $x^{k_c}$ from the shared memory
4:    Compute a subgradient estimate $g^{k_c}$, local to $k_c$
5:    Write, with a lock, to the shared memory vector partition $u_{i^{k_c}} = m u_{i^{k_c}} + g_{i^{k_c}}^{k_c}$
6:    Update, with a lock, to the shared memory vector partition $x_{i^{k_c}} = x_{i^{k_c}} - (1-m)\gamma^{k_c} u_{i^{k_c}}$
7:    $k_c = k_c + 1$
8: **end while**

---

## 3 ANALYSIS

For the discrete time probabilistic model of computation introduced in Cannelli et al., we must present the basic requirements that must hold across cores. In particular, it is reasonable to expect that if some core is entirely faulty, or exponentially decelerates in its computation, convergence should not be expected to be attained. Otherwise we wish to make the probabilistic assumption

---

[1]The proposed method will work even if only a CREW access is available because of partitioning of variables.

governing the asynchronous update scheduling as general as possible in allowing for a variety of possible architectures.

The details of the probabilistic assumptions are technical and left to the Supplementary Material. It can be verified that the stochastic approxmation framework discussed in the next section detailing the convergence satisfies these assumptions.

We have the standard assumption about the stochastic sub-gradient estimates. These assumptions hold under the standard stochastic gradient approach wherein one samples some subset $\xi \subseteq \{1, ..., M\}$ of mini-batches uniformly from the set of size $|\xi|$ subsets of $\{1, ..., M\}$, done independently at each iteration. This results in independent noise at each iteration being applied to the stochastic subgradient term. From these mini-batches $\xi$, a subgradient is taken for each $j \in \xi$ and averaged.

**Assumption 3.1.** *The stochastic subgradient estimates $g(x, \xi)$ satisfy,*

1. $\mathbb{E}_\xi \left[ g(x; \xi) \right] \in \partial f(x) + \beta(x)$

2. $\mathbb{E}_\xi \left[ dist(g(x; \xi), \partial f(x))^2 \right] \leq \sigma^2$

3. $\|g(x; \xi)\| \leq B_g$ *w.p.* 1.

where $\beta(x)$ defines a bias term that is zero if $f(\cdot)$ is continuously differentiable at $x$.

We provide some more details on the "global" model of asynchronous stochastic updating in the Supplementary material.

## 3.1 CONTINUOUS TIME MODEL AND STOCHASTIC PROCESS

In this section, we shall redefine the algorithm and its associated model presented in the previous section in a framework appropriate for analysis from the stochastic approximation perspective.

Consider the Algorithm described as such, for data block $i$ with respective iteration $k$,

$$x_{k+1,i} = x_{k,i} + (1 - m)\gamma^{k,i} \sum_{j=1}^{k} m^{k-j} Y_{j,i} \tag{2}$$

where $Y_{j,i}$ is the estimate of the partial subgradient with respect to block variables indexed by $i$ at local iteration $j$.

In the context of Algorithm 1, the step size is defined to be the subsequence $\{\gamma^{k,i}\} = \{\gamma^{\nu(c(i),l)} : i = i^l\}$ where $l$ is the iteration index for the core corresponding to block $i$. Thus it takes the subsequence of $\gamma^k$ for which $i^k = i$ is the block of variables being modified.

The step $Y_{k,i}$ corresponding to $g(xk, \xi)$ satisfies,
$$Y_{k,i} = g_i((x_{k-[d_i^k]_1,1}, ..., x_{k-[d_i^k]_j,j}, ..., x_{k-[d_i^k]_n,n})) + \beta_{k,i} + \delta M_{k,i}.$$
We denote $g_i(x)$ to denote a selection of some element of the subgradient, with respect to block $i$, of $f(x)$. The quantity $\delta M_{k,i}$ represents a Martingale difference, satisfying $\delta M_{k,i} = M_{k+1,i} - M_{k,i}$ for some Martingale $M_k$, a sequence of random variables which satisfies $\mathbb{E}[M_{k,i}] < \infty$ and $\mathbb{E}[M_{k+1,i}|M_{j,i}, j \leq k] = M_{k,i}$ with probability 1 for all $k$. It holds that $\mathbb{E}[|M_{k,i}|^2] < \infty$ and $\mathbb{E}[M_{k+1,i} - M_{k,i}][M_{j+1,i} - M_{j,i}]' = 0$. Finally, it holds that $\mathbb{E}_{k,i}[\delta M_{k,i}] = 0$. These are standard conditions implied by the sampling procedure in stochastic gradient methods, introduced by the original Robbins-Monro method (Robbins & Monro, 1985).

In Stochastic Approximation, the standard approach is to formulate a dynamic system or differential inclusion that the sequence of iterates approaches asymptotically. For this reason, we introduce real time into the model of asynchronous computation, looking at the actual time elapsed between iterations for each block $i$.

Define $\delta\tau_{k,i}$ to be the real elapsed time between the $k$-th and $k + 1$-st iteration for block $i$. We let $T_{k,i} = \sum_{j=0}^{k-1} \delta\tau_{j,i}$ and define for $\sigma \geq 0$, $p_l(\sigma) = \min\{j : T_{j,i} \geq \sigma\}$ the first iteration at or after $\sigma$.

We assume now that the step-size sequence comes from an underlying real function, i.e.,

$$\gamma^{k,i} = \frac{1}{\delta\tau_{k,i}} \int_{T_{k,i}}^{T_{k,i}+\delta\tau_{k,i}} \gamma(s)ds$$

satisfying

$$\int_0^\infty \gamma(s)ds = \infty, \text{ where } 0 < \gamma(s) \to 0 \text{ as } s \to \infty,$$
$$\text{There are } T(s) \to \infty \text{ as } s \to \infty \text{ such that } \lim_{s\to\infty} \sup_{0 \le t \le T(s)} \left| \frac{\gamma(s)}{\gamma(s+t)} - 1 \right| = 0 \tag{3}$$

We now define new $\sigma$-algebras $\mathcal{F}_{k,i}$ and $\mathcal{F}_{k,i}^+$ defined to measure the random variables

$$\{\{x_0\}, \{Y_{j-1,i} : j, i \text{ with } T_{j,i} < T_{k+1,i}\}, \{T_{j,i} : j, i \text{ with } T_{j,i} \le T_{k+1,i}\}\}, \text{ and,}$$
$$\{\{x_0\}, \{Y_{j-1,i} : j, i \text{ with } T_{j,i} \le T_{k+1,i}\}, \{T_{j,i} : j, i \text{ with } T_{j,i} \le T_{k+1,i}\}\},$$

indicating the set of events up to, and up to and including the computed noisy update at $k$, respectively.

Note that each of these constructions is still consistent with a core updating different blocks at random, with $\delta\tau_{k,i}$ arising from an underlying distribution for $\delta\tau_{k,c(i)}$.

Let us relate these $\sigma$-algebras to those in the previous section. Note that this takes subsets of random variables $(i^k, d^k, \xi^k)$ for which $k$ is such that $i^k$ is $i$ (in the original notation of $k$). The form of $Y_{k,i}$ defined above incorporates the random variable $d^k$ and $i^k$, as in which components are updated and the age of the information used by where the subgradient is evaluated, as well as $\xi^k$ by the presence of the Martingale difference noise.

For any sequence $Z_{k,i}$ we write $Z_{k,i}^\sigma = Z_{p_i(\sigma)+k,i}$, where $p_i(\sigma)$ is the least integer greater than or equal to $\sigma$. Thus, let $\delta\tau_{k,i}^\sigma$ denote the inter-update times for block $i$ starting at the first update at or after $\sigma$, and $\gamma_{k,i}^\sigma$ the associated step sizes.

Now let $x_{0,i}^\sigma = x_{p_i(\sigma),i}$ and for $k \ge 0$, $x_{k+1,i}^\sigma = x_{k,i}^\sigma + (1-m)\gamma_{k,i}^\sigma \sum_{j=1}^k m^{k-j} Y_{j,i}^\sigma$.

We consider $t_{k,i}^\sigma = \sum_{j=0}^{k-1} \gamma_{j,i}^\sigma$ and $\tau_{k,i}^\sigma = \sum_{j=0}^{k-1} \gamma_{j,i}^\sigma \delta\tau_{j,i}^\sigma$.

We introduce piecewise constant interpolations of the vectors in real-time given by,

$$x_i^\sigma(t) = x_{k,i}^\sigma, \quad \hat{x}_i^\sigma(t) = \hat{x}_{k,i}^\sigma, \quad N_i^\sigma(t) = t_{k,i}^\sigma, \quad t \in [\tau_{k,i}^\sigma, \tau_{k+1,i}^\sigma)$$

and $\tau_i^\sigma(t) = \tau_{k,i}^\sigma$ for $t \in [t_{k,i}^\sigma, t_{k+1,i}^\sigma]$. We also have,

$$N_i^\sigma(\tau_i^\sigma(t)) = t_{k,i}^\sigma, \, t \in [t_{k,i}^\sigma, t_{k+1,i}^\sigma], \, x_i^\sigma(t) = \hat{x}_i^\sigma(\tau_i^\sigma(t)), \, \hat{x}_i^\sigma(t) = x_i^\sigma(N_i^\sigma(t))$$

Now we detail the assumptions on the real delay times. These ensure that the real-time delays do not grow without bound, either on average, or on relevantly substantial probability mass. Intuitively, this means that it is highly unlikely that any core deccelerates exponentially in its computation speed.

**Assumption 3.2.** *It holds that $\{\delta\tau_{k,i}^\sigma; k, i\}$ is uniformly integrable.*

**Assumption 3.3.** *There exists a function $u_{k+1,i}^\sigma$ and random variables $\Delta_{k+1,i}^{\sigma,+}$ and a random sequence $\{\psi_{k+1,i}^\sigma\}$ such that*

$$\mathbb{E}_{k,i}^+[\delta\tau_{k+1,i}^\sigma] = u_{k+1,i}^\sigma(\hat{x}_i^\sigma(\tau_{k+1,i}^\sigma - \Delta_{k+1,i}^{\sigma,+}), \psi_{k+1,i}^\sigma)$$

*and there is a $\bar{u}$ such that for any compact set $A$,*

$$\lim_{m,k,\sigma} \frac{1}{m} \sum_{j=k}^{k+m-1} E_{k,i}^\sigma[u_{j,i}^\sigma(x, \psi_{k+1,i}^\sigma) - \bar{u}_i(x)]I_{\{\psi_{k+1,i}^\sigma \in A\}} = 0$$

**Assumption 3.4.** *It holds that,*

$$\lim_{m,k,\sigma} \frac{1}{m} \sum_{j=k}^{k+m-1} \mathbb{E}_{k,i}^\sigma[\beta_{j,i}^\sigma] = 0 \tag{4}$$

This assumption holds if, e.g., the set of $x$ such that $f(\cdot)$ is not continuously differentiable at $x$ is of measure zero, which is the case for objectives of every DNN architecture the authors are aware of.

### 3.2 CONVERGENCE

As mentioned earlier, the primary goal of the previous section is to define a stochastic process that approximates some real-time process asymptotically, with this real-time process defined by dynamics for which at the limit the path converges to a stationary point. In particular, we shall see that the process defined for the iterate time scale approximates the path of a differential inclusion,

$$\dot{x}_i(t) \in \partial_i f(x(t)) \tag{5}$$

and we shall see that this path defines stationary points of $f(\cdot)$.

We must define the notion of an invariant set for a differential inclusion (DI).

**Definition 3.1.** *A set $\Lambda \subset \mathbb{R}^n$ is an invariant set for a DI $\dot{x} \in g(x)$ if for all $x_0 \in \Lambda$, there is a solution $x(t)$, $-\infty < t < \infty$ that lies entirely in $\Lambda$ and satisfies $x(0) = x_0$.*

Now we state our main result. Its complete proof can be found in the Supplementary Material.

**Theorem 3.1.** *Let all the stated Assumptions hold.*

*Then, the following system of differential inclusions,*

$$\tau_i(t) = \int_0^t \bar{u}_i(\hat{x}(\tau_i(s)))ds, \quad \dot{x}_i(t) \in \partial_i f(\hat{x}(\tau_i(t))), \quad \dot{\hat{x}}_i(t)\bar{u}_i(\hat{x}) \in \partial_i f(\hat{x}(t)) \tag{6}$$

*holds for any $\bar{u}$ satisfying 3.3. On large intervals $[0, T]$, $\hat{x}^\sigma(\cdot)$ spends nearly all of its time, with the fraction going to one as $T \to \infty$ and $\sigma \to \infty$ in a small neighborhood of a bounded invariant set of* (5).

This Theorem shows weak convergence. The extension to convergence with probability one is straightforward and described in the Supplementary material.

#### 3.2.1 PROPERTIES OF THE LIMIT POINT

Finally, we wish to characterize the properties of this invariant set. From Corollary 5.11 (Davis et al., 2018), we can conclude that problems arising in training of deep neural network architectures, wherein $f(x) = l(y_j, a_L)$ with $l(\cdot)$ one of several standard loss functions, including logistic or Hinge loss, and $a_i = \rho_i(V_i(x)a_{i-1})$ or $i = 1, ..., L$ layers, are activation functions, which are piece-wise defined to be $\log x$, $e^x$, $\max(0, x)$ or $\log(1 + e^x)$, are such that their set of invariants $\{x^*\}$ for its associated differential inclusion satisfies $0 \in \partial f(x^*)$, and furthermore the values $f(x^k)$ for any iterative algorithm generating $\{x^k\}$ such that $x^k \to x^*$, an invariant of $f(x)$, converge.

Note that the differential inclusions defined above ensure asymptotic convergence to block-wise stationarity, i.e., $0 \in \partial_i f(x)$ for all $i$. It is clear, however, that every stationary point is also block-wise stationary, i.e., that $0 \in \partial f(x)$ implies $0 \in \partial_i f(x)$ for all $i$. In practice, the set of block-wise stationary points which are not stationary is not large. One can alternatively consider a variant of the algorithm wherein every core updates the entire vector (thus there is no block partitioning) but locks the shared memory whenever it either reads of writes from it. The same analysis applies to such a procedure. In particular, this amounts to $i^k = \{1, ..., n\}$ for all $k$ and every limit of $x^\sigma(t)$ as either $\sigma \to \infty$ or $t \to \infty$ is a critical point of $f(x)$ and, with probability one, asymptotically the algorithm converges to a critical point of $f(x)$ (i.e., $x$ such that $0 \in \partial f(x)$).

## 4 NUMERICAL RESULTS

**Methods.** We describe an experimental evaluation comparing the following algorithms:

**WIASSM: W**rite **In**consistent **A**synchronous **S**tochastic **S**ubgradient **M**ethod with lock-free read and updates of $x_{k,i}$. This procedure applied to smooth strongly-convex and smooth nonconvex $f(x)$ is known as HogWild! in Recht et al. (2011) and AsySG-incon in Lian et al. (2015), respectively, in the literature. Convergence analysis of HogWild! and AsySG-incon additionally required sparsity of $x$. They have no provable convergence guarantee for nonsmooth nonconvex models.

**WCASSM: W**rite **C**onsistent **A**synchronous **s**tochastic **s**ubgradient **m**ethod. WCASSM differs from WIASSM in its use of locks to update $x_{k,i}$ to make consistent writes. AsySG-con in Lian et al. (2015) is its counterpart for smooth nonconvex $f(x)$ and sparse $x$.

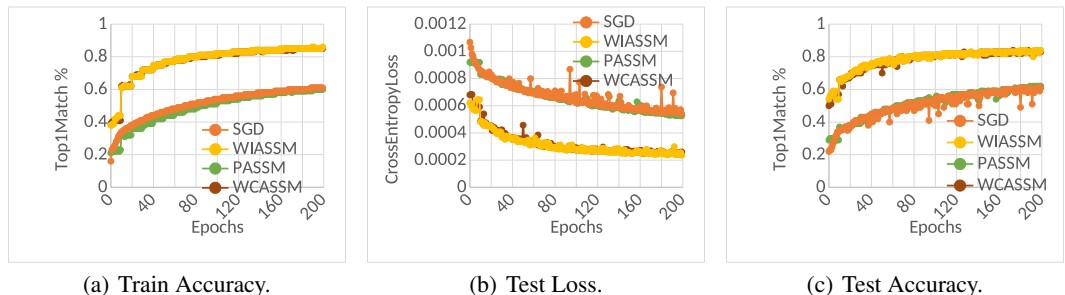

| (a) Train Accuracy. | (b) Test Loss. | (c) Test Accuracy. |

Figure 1: We plotted the train accuracy and generalization (test) loss and accuracy trajectories for the methods. SGD runs a single process, whereas the asynchronous methods run 10 concurrent processes. In this set of experiments we have no momentum correction. The WIASSM and WCASSM demonstrate better convergence per epoch compared to PASSM. Note that, the single process executing SGD iterations has a better opportunity to use CUDA threads as there is no concurrent use of GPUs by multiple processes. The per epoch performance of PASSM matches that of SGD inferring that amount of subgradient updates are almost identical: in the former it is done collectively by all the concurrent processes accessing disjoint set of tensors, whereas, in the latter it is done by a single process using comparable amount of parallization.

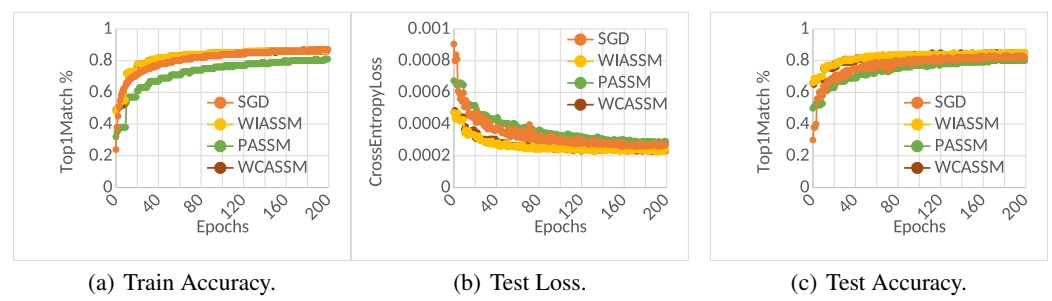

| (a) Train Accuracy. | (b) Test Loss. | (c) Test Accuracy. |

Figure 2: Same setting as in Fig 1. We used a momentum = 0.9. It can be observed that with momentum correction the convergence of PASSM improves significantly. Mitliagkas et al. Mitliagkas et al. (2016) experimentally showed that the degree of asynchrony directly relates to momentum; our experiments show that the relative gain in terms of convergence per epoch by momentum correction is better for PASSM that exhibits more asynchrony compared to WCASSM, which uses locks for write consistency.

**PASSM:** The presented **P**artitioned **A**synchronous **S**tochastic **S**ubgradient **M**ethod. We read as well as update $x_{k,i}$ lock-free asynchronously.

**SGD:** Sequential data-parallel **S**tochastic **G**radient **D**escent method.

**Hyper-parameters.** For each of the methods, we adopt a decreasing step size strategy $\gamma^{k,i} = (\alpha^j \times \gamma)/\sqrt{k}$, where $\alpha^j > 0$ is a constant for the $j^{th}$ processing core. $\gamma$ is fixed initially. In each of the methods we use an $L2$ penalty in the form of a weight-decay of 0.0005. Additionally, we introduced an $L1$ penalty of 0.0001 that simply gets added to the gradients after it has been put through the $L2$ penalty. In accordance with the theory, we explored the effect of momentum correction: we have two sets of benchmarks, one without momentum and another with a constant momentum of 0.9 while checking the convergence with epochs. In all of the above methods we load the datasets in mini-batches of size 64. We keep the hyper-parameters, in particular, learning rate and mini-batch-size, identical across methods for the sake of statistical fairness. In a shared-memory setting, there is not much to exploit on the front of saving on communication cost as some existing works do Goyal et al. (2017); and the computing units, see the system setting and the implementation below, are anyway utilized to their entirety by way of efficient data-parallelization.

**Dataset and Networks.** We used CIFAR10 data set of RGB images Krizhevsky (2009). It contains 50000 labeled images for training and 10000 labeled images for testing. We trained a well known

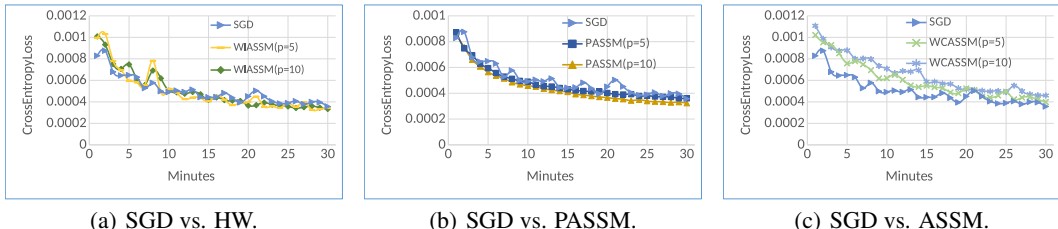

(a) SGD vs. HW.     (b) SGD vs. PASSM.     (c) SGD vs. ASSM.

Figure 3: This set of figures presents the train-loss trajectory against time (in minutes) while comparing asynchronous methods – running 5 and 10 concurrent processes – with sequential SGD. We used momentum = 0.9 in each of them. As described, a separate concurrent process keeps on saving a snapshot of the shared model at an interval of 1 minute, simultaneously with the training processes. Firstly, it can be observed that the convergence of PASSM is faster compared to the other two asynchronous methods for identical number of processes. This can be understood in terms of block partitioning the model across processes: it helps reducing the synchronization cost and thereby potentially speeds up the data processing per unit time. Furthermore, we clearly gain in terms of convergence per unit time when we increase the number of processes in PASSM. In contrast, we note that the use of locks by WCASSM actually slows it down when we increase the number of processes. This set of experiments demonstrate that PASSM has better convergence with respect to wall-clock time in addition to the scalability with parallel resources.

CNN model Resnet18 He et al. (2016). ResNet18 has a blocked architecture – of residual blocks – totaling 18 convolution layers. Each residual block is followed by a ReLU activation causing non-linearity. Evidently, training of this neural network offers general nonsmooth nonconvex optimization problems.

**System Specification.** We benchmarked the implementations on a NUMA workstation – 2 sockets, 10 cores apiece, running at 2.4GHz (Intel(R) Xeon(R) E5- 2640), HT enabled 40 logical cores, Linux 4.18.0-0.bpo.1-amd64 (Debian 9) – containing 4 Nvidia GeForce GTX 1080 GPUs. For a fair evaluation of scalability with cores, we bind the processes restricting their computations – in particular, the cost of data load – to individual CPU cores. In this setting, to evaluate the scalability with respect to wall-clock-time by increasing the availability of parallel resources, we run the experiments with 5 and 10 processes, which do not migrate across CPU sockets. For evaluation with respect to time, we employed a separate concurrent process that keeps on saving a snapshot of the shared model at an interval of 1 minute.

**Asynchronous Implementation.** We implemented the asynchronous methods using the open-source Pytorch library Paszke et al. (2017) and the multi-processing framework of Python. Given the multi-GPU environment, which could excellently exploit data-parallel computation, therefore, we used the `nn.DataParallel()` module of Pytorch library. Thereby, a CNN instance is centrally allocated on one of the GPUs and computations – forward pass to compute the model over a computation graph and backward pass to compute the sub-gradients thereon – employ peer GPUs by replicating the central instance on them for each mini-batch in a data-parallel way. Effectively, the computation of stochastic subgradients happen over each GPU and they are summed and added to the central instance. Note that, this way of implementation exploits parallel resources while effectively simulating a shared-memory asynchronous environment.

**Model Partitioning.** Unlike PASSM, the methods WIASSM, WCASSM and SGD do not partition the model and compute the stochastic subgradients over the entire computation graph of a CNN via backpropagation provided by the autograd module of Pytorch. PASSM partitions the list of leaves, which are tensors corresponding to the weights and biases, of the computation graph into blocks. While computing the stochastic subgradients with respect to a block, we switch off the `requires_grad` flag of the tensors corresponding to other blocks during backpropagation. This particular implementation component results in some savings in stochastic sub-gradient computation with respect layers relatively closer to the output. Keeping this in view, we assigned blocks containing $s_i \geq \lceil L/p \rceil$ parameter components, where $L$ is the model size and $p$ is the number of processes, to the processes $P_i$ computing stochastic sub-gradients corresponding to layers closer to output. Whereas, the process that computes sub-gradient of the layer-block closest to the input is assigned a block containing less than $\lfloor L/p \rfloor$ parameter components. The assignments $s_i$ aim to balance computation

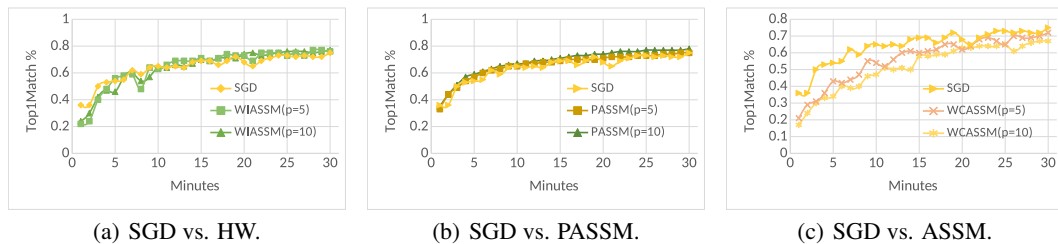

(a) SGD vs. HW.   (b) SGD vs. PASSM.   (c) SGD vs. ASSM.

Figure 4: Same setting as in the Fig 3, momentum = 0.9. We plotted test-accuracy in terms of Top1 correct match % vs time (in minutes). In can be observed that PASSM offers faster convergence per unit time in accuracy as well compared to the other two asynchronous methods.

load, however, it varies across layers depending on the size of the assigned leaves in terms of parameter component. Nevertheless, a blocked architecture such as ResNet does not allow much scope of computation-cost saving on this count: we observed an insignificant difference in average processing time for the same number of epochs irrespective of switching off the `requires_grad` flag. Notice that, this is *not a model parallelization* and the stochastic subgradient computation with respect to a leaf depends on the computation path leading to the output. Irrespective of partitioning the model, the multi-GPU-based data-parallel implementation utilizes replication and data partitioning over GPUs while processing a mini-batch.

The experimental observations are described in Figures 1, 2, 3, and 4.

**Summary.** The block partitioning design of PASSM has its efficiency in the following: 1) it reduces the cost of optimization per process, since the parameter is partitioned. Note that, in case of neural networks, where backpropagation processes almost the entire computation graph irrespective of the location of the leaf, in particular in a blocked architecture such as ResNet, PASSM clearly misses out saving subgradient computation cost by way of parallelization; it can be significantly better if the subgradients with respect to the blocks could be computed independently; and 2) reduces memory traffic and potential write conflicts between processors which we observe in terms of better convergence per unit time. And finally, it is pertinent to highlight that we also observed that momentum correction improves the convergence per epoch of the block partitioning approach whose performance was way lower if we did not use it.

## 5 DISCUSSION AND CONCLUSION

In this paper we analyzed the convergence theory of asynchronous stochastic subgradient descent. We found that the state of the art probabilistic model on asynchronous parallel architecture applied to the stochastic subgradient method, with and without the use of momentum, is consistent with standard theory in stochastic approximation and asymptotic convergence with probability one holds for the method under the most general setting of asynchrony.

We presented numerical results that indicate some possible performance variabilities in three types of asynchrony: block partitioning inconsistent read (for which the above convergence theory applies), full-variable-update consistent write (for which the above convergence theory also applies), and full-variable-update inconsistent read/write (for which no convergence theory exists).

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

# 6 APPENDIX A: GLOBAL SHARED MEMORY ASYNCHRONOUS STOCHASTIC SUBGRADIENT MODEL

Here we give a few more details describing the relation of the probabilistic model of asynchrony to the underlying hardware properties, as modeled in Cannelli et al..

In this section, we present $k$ as a *global* counter, indicating sequential updates of any block among the variables.

In iteration $k$, the updated iterate $x_{i^k}^{k+1}$ depends on a random vector $\zeta^k \triangleq (i^k, d^k, \xi^k)$. The distribution of $\zeta^k$ depends on the underlying scheduling or message passing protocol. We use the following formulation, which applies to a variety of architectures.

Let $\zeta^{0:t} \triangleq (\zeta^0, \zeta^1, ..., \zeta^t)$ be the stochastic process representing the evolution of the blocks and minibatches used, as well as the iterate delays. The $\sigma$-algebra $\mathcal{F}$ is obtained as follows. Let the cylinder $C^k(\zeta^{0:t}) \triangleq \{\omega \in \Omega : \omega^{0:k} = \zeta^{0:t}\}$ and define $\mathcal{F}^k \triangleq \sigma(C^k)$ and $\mathcal{F} \triangleq \sigma(\cup_{t=0}^\infty C^t)$ the cylinder $\sigma$-algebra on $\Omega$.

Consider the conditional distribution of $\zeta^{k+1}$ given $\zeta^{0:k}$,

$$\mathbb{P}(\zeta^{k+1}|\zeta^{0:k}) = \frac{\mathbb{P}(C^{k+1}(\zeta^{0:k+1})}{\mathbb{P}(C^k(\zeta^{0:k}))},$$

we have the following assumptions on the probabilities of block selection and the delays,

**Assumption 6.1.** *The random variables $\zeta^k$ satisfy,*

1. *There exists a $\delta$ such that $d_j^k \le \delta$ for all $j$ and $k$. Thus each $d_j^k \in \mathcal{D} \triangleq \{0, ..., \delta\}^n$.*

2. *For all $i$ and $\zeta^{0:k-1}$ such that $p_{\zeta^{0:k-1}}(\zeta^{0:k-1}) > 0$, it holds that,*

$$\sum_{d \in \mathcal{D}} \mathbb{P}((i, d, \xi)|\zeta^{0:k-1}) \ge p_{min}$$

   *for some $p_{min} > 0$.*

3. *It holds that,*

$$\mathbb{P}\left(\left\{\zeta \in \Omega : \liminf_{k \to \infty} \mathbb{P}(\zeta|\zeta^{0:k-1}) > 0\right\}\right) = 1$$

The first condition indicates that there is some maximum possible delay in the vectors, that each element of $x$ used in the computation of $x_{i^k}^{k+1}$ is not too old. The second is an irreducibility condition that there is a positive probability for any block or minibatch to be chosen, given any state of previous realizations of $\{\zeta^k\}$. The last assumption indicates that the set of events in $\Omega$ that asymptotically go to zero in conditional probability are of measure zero.

In order to enforce global convergence, we wish to use a diminishing step-size. However, at the same time, as synchronization is to be avoided, there must not be a global counter indicating the rate of decrease of the step-size. In particular, each core will have its own local step size $\gamma^{\nu(c^k, k)}$ where $c^k$ is the core, and, defining the random variable $Z^k$ as the component of $\{1, ..., \bar{c}\}$ that is active at iteration $k$, the random variable denoting the number of updates performed by core $c^k$, denoted by $\nu(k)$ is given by $\nu(k) \triangleq \sum_{j=0}^k I(Z^j = c^k)$.

In addition, noting that it has been observed that in practice, partitioning variable blocks across cores is more efficient than allowing every processor to have the ability to choose across every variable block (Liu & Wright, 2015). Thus we partition the blocks of variables across cores. We can thus denote $c^k$ as being defined uniquely by $i^k$, the block variable index updated at iteration $k$.

Note that $\liminf_{k \to \infty} \frac{\gamma^{\nu(c^k, k)}}{k} = 0$ in probability is implied by

$$\sum_{i \in c^k, d \in \mathcal{D}, \xi \subseteq \{1, ..., M\}} Pr((i, d, \xi)|\zeta^{0:k-1})) \to 0$$

for some subsequence, which is antithetical to Assumption 3.1, Part 2. Thus, note that the stepsizes $\gamma^{\nu(c^k,k)}$ satisfy, where the limit of the sequence is taken in probability,

$$\liminf_{k\to\infty} \frac{\gamma^{\nu(c^k,k)}}{k} > 0, \tag{7}$$

which is an assumption for the analysis of asynchronous parallel algorithms in Borkar (2008).

We are now ready to present Algorithm 2. This is presented from the "global" iteration counter perspective.

---

**Algorithm 2** Asynchronous Stochastic Subgradient Method

$\qquad\qquad$ **Input:** $x_0$.
1: **while** Not converged and $k < k_{\max}$ **do**
2: $\qquad$ Having realized $\zeta^{0:k-1}$, sample $\{\zeta^k = (i^k, d^k, \xi^k)\}|\zeta^{0:k-1}\}$.
3: $\qquad$ Update $u_{i^k} = m u_{i^k} + g((x_1^{d_1^k}, x_2^{d_2^k}, ..., x_n^{d_n^k}), \xi^k)$
4: $\qquad$ Update $x_{i^k}^{k+1} = x_{i^k}^k - (1-m)\gamma^{\nu(k)} u_{i^k}$
5: $\qquad$ Set $k = k+1$
6: **end while**

---

## 7 APPENDIX B: PRELIMINARY ASSUMPTIONS AND LEMMAS

**Lemma 7.1.** *It holds that* $\{Y_{k,i}, Y_{k,i}^\sigma; k, i\}$ *is uniformly integrable. Thus, so is* $\left\{\sum_{j=1}^k m^{k-j} Y_{j,i}, \sum_{j=1}^k m^{k-j} Y_{j,i}^\sigma; k, i\right\}$

*Proof.* Uniform integrability of $\{Y_{k,i}, Y_{k,i}^\sigma; k, i\}$ follows from Assumption 3.2, part 3. The uniform integrability of $\left\{\sum_{j=1}^k m^{k-j} Y_{j,i}, \sum_{j=1}^k m^{k-j} Y_{j,i}^\sigma; k, i\right\}$ follows from $0 < m < 1$ and the fact that a geometric sum of a uniformly integrable sequence is uniformly integrable. $\qquad\square$

**Lemma 7.2.** *It holds that, for any $K > 0$, and all $l$,*

$$\sup_{k<K} \sum_{j=k-[d_i^k]_l}^k \gamma_{j,i}^\sigma \to 0$$

*in probability as $\sigma \to \infty$.*

*Proof.* As $\sigma \to \infty$, by the definition of $\gamma_{k,i}^\sigma$, $\gamma_{k,i}^\sigma \to 0$ and since by Assumption 3.1 $\max d_i^k \le \delta$, for all $k < K$, $\sum_{j=k-[d_i^k]_l}^k \gamma_{j,i}^\sigma \le \delta\gamma_{k-\delta,i}^\sigma \to 0$. $\qquad\square$

Now we define some terminology arising in the theory of weak convergence. We present a result indicating sufficient conditions for a property called tightness.

**Theorem 7.1.** *(Kushner & Yin, 2003, Theorem 7.3.3) Consider a sequence of processes $\{A_k(\cdot)\}$ with paths in $D(-\infty, \infty)$ such that for all $\delta > 0$ and each $t$ in a dense set of $(-\infty, \infty)$ there is a compact set $K_{\delta,t}$ such that,*

$$\inf_n \mathbb{P}\left[A_n(t)| \in K_{\delta,t}\right] \ge 1 - \delta,$$

*and for any $T > 0$,*

$$\lim_{\delta\to 0} \limsup_n \sup_{|\tau|\le T} \sup_{s\le\delta} \mathbb{E}\left[\min\left[|A_n(\tau+s) - A_n(\tau)|, 1\right]\right] = 0$$

*then $\{A_n(\cdot)\}$ is tight in $D(-\infty, \infty)$.*

If a sequence is tight then every weak sense limit process is also a continuous time process. We say that $A_k(t)$ *converges weakly* to $A$ if,

$$\mathbb{E}\left[F(A_k(t))\right] \to \mathbb{E}\left[F(A(t))\right]$$

for any bounded and continuous real-valued function $F(\cdot)$ on $\mathbb{R}^n$.

Weak convergence is defined in terms of the Skorohod topology, a technical topology weaker than the topology of uniform convergence on bounded intervals, defined in Billingsley (1968). Convergence of a function $f_n(\cdot)$ to $f(\cdot)$ in the Skorohod topology is equivalent to uniform convergence on each bounded time interval. We denote by $D^j[0, \infty)$ the $j$-fold product space of real-valued functions on the interval $[0, \infty)$ that are right continuous with left-hand limits, with the Skorohod topology. It is a complete and separable metric space.

Much of the proof of the main Theorem can be taken from the analagous result in Chapter 12 of Kushner & Yin (2003), which considers a particular model of asynchronous stochastic approximation. As we introduced a slightly different model from the literature, some of the details of the procedure are now different, and furthermore we introduced momentum to the algorithm, and thus in the next section we indicate how to treat the distinctions in the proof and show that the result still holds.

# 8    APPENDIX C: PROOF OF THEOREM 1

By Theorem 8.6, Chapter 3 in Ethier & Kurtz (2009) a sufficient condition for tightness of a sequence $\{A_n(\cdot)\}$ is that for each $\delta > 0$ and each $t$ in a dense set in $(-\infty, \infty)$, there is a compact set $K_{\delta,t}$ such that $\inf_n \mathbb{P}[A_n(t) \in K_{\delta,t}] \geq 1 - \delta$ and for each positive $T$, $\lim_{\delta \to 0} \limsup_n \sup_{|\tau| \leq T, \, s \leq \delta} \mathbb{E}\left[\|A_n(\tau + s) - A_n(\tau)\|\right] = 0$.

Now since $Y_{k,i}$ is uniformly bounded, and $Y_{k,i}^{\sigma}(\cdot)$ is its interpolation with jumps only at $t$ being equal to some $T_{k,i}$, it holds that for all $i$,

$$\lim_{\delta \to 0} \limsup_{\sigma} \mathbb{P}\left[\sup_{t \leq T, \, s \leq \delta} |Y_{k,i}^{\sigma}(t + s) - Y_{k,i}^{\sigma}(t)| \geq \eta\right] = 0$$

and so by the definition of the algorithm,

$$\lim_{\delta \to 0} \limsup_{\sigma} \mathbb{P}\left[\sup_{t \leq T, \, s \leq \delta} |x_{k,i}^{\sigma}(t + s) - x_{k,i}^{\sigma}(t)| \geq \eta\right] = 0$$

which implies,

$$\lim_{\delta \to 0} \limsup_{\sigma} \mathbb{E}\left[\sup_{t \leq T, \, s \leq \delta} |x_{k,i}^{\sigma}(t + s) - x_{k,i}^{\sigma}(t)|\right] = 0$$

and the same argument implies tightness for $\{\tau_i^{\sigma}(\cdot), N_i^{\sigma}(\cdot)\}$ by the uniform boundedness of $\{\delta\tau_{i,k}^{\sigma}\}$ and bounded, decreasing $\gamma_{k,i}^{\sigma}$ and positive $u_{k,i}^{\sigma}(x, \psi_{k+1,i}^{\sigma})$, along with Assumption 3.4. Lipschitz continuity follows from the properties of the interpolation functions. Specifically, the Lipschitz constant of $x_i^{\sigma}(\cdot)$ is $B_g$.

All of these together imply tightness of $\hat{x}_i^{\sigma}(\cdot)$ as well. Thus,

$$\{x_i^{\sigma}(\cdot), \tau_i^{\sigma}(\cdot), \hat{x}_i^{\sigma}(\cdot), N_i^{\sigma}(\cdot); \sigma\}$$

is tight in $D^{4n}[0, \infty)$. This implies the Lipschitz continuity of the subsequence limits with probability one, which exist in the weak sense by Prohorov's Theorem, Theorems 6.1 and 6.2 (Billingsley, 2013).

As $\sigma \to \infty$ we denote the weakly convergent subsequence's weak sense limits by,

$$(x_i(\cdot), \tau_i(\cdot), \hat{x}_i(\cdot), N_i(\cdot))$$

Note that,

$$x_i(t) = \hat{x}_i(\tau_i(t)),$$
$$\hat{x}_i(t) = x_i(N_i(t)),$$
$$N_i(\tau_i(t)) = t.$$

For more details, see the proof of Kushner & Yin (2003, Theorem 8.2.1).

Let,

$$M_i^\sigma(t) = \sum_{k=0}^{k=p(\sigma)} (1-m)\delta\tau_{k,i} \left( \sum_{j=0}^k m^j \delta M_{k-j,i}^\sigma \right)$$

$$\tilde{G}_i^\sigma(t) = \sum_{k=0}^{k=p(\sigma)} \delta\tau_{k,i} \left[ (1-m) \sum_{j=0}^k m^j g_i((x_{k-j-[d_i^{k-j}]_1,1}^\sigma(t), ..., \right.$$

$$\left. x_{k-j-[d_i^{k-j}]_j,j}^\sigma(t), ..., x_{k-j-[d_i^{k-j}]_N,N}^\sigma))(t) - g_i(\hat{x}_i^\sigma(t)) \right]$$

$$\bar{G}_i^\sigma(t) = \sum_{k=0}^{k=p(\sigma)} \delta\tau_{k,i} g_i(\hat{x}^\sigma(t))$$

$$B_i^\sigma(t) = \sum_{k=0}^{\rho(\sigma)} (1-m)\delta\tau_{k,i} \left( \sum_{j=0}^k m^j \beta_{k-j,i}^\sigma \right)$$

$$W_i^\sigma(t) = \hat{x}_i^\sigma(\tau_i^\sigma(t)) - x_{i,0}^\sigma - \bar{G}_i^\sigma(t) = \tilde{G}_i^\sigma(t) + M_i^\sigma(t)$$

Now for any bounded continuous and real-valued function $h(\cdot)$, an arbitrary integer $p$, and $t$ and $\tau$, and $s_j \geq t$ real, we have

$$\mathbb{E}\left[ h(\tau_i^\sigma(s_j), \hat{x}^\sigma(\tau_i^\sigma(s_j)) (W_i^\sigma(t+\tau) - W_i^\sigma(t)) \right]$$

$$-\mathbb{E}\left[ h(\tau_i^\sigma(s_j), \hat{x}^\sigma(\tau_i^\sigma(s_j)) \left( \tilde{G}_i^\sigma(t+\tau) - \tilde{G}_i^\sigma(t) \right) \right]$$

$$-\mathbb{E}\left[ h(\tau_i^\sigma(s_j), \hat{x}^\sigma(\tau_i^\sigma(s_j)) (M_i^\sigma(t+\tau) - M_i^\sigma(t)) \right]$$

$$-\mathbb{E}[ h(\tau_i^\sigma(s_j), \hat{x}^\sigma(\tau_i^\sigma(s_j)))(B_i^\sigma(t+\tau) - B_i^\sigma(t))] = 0.$$

Now the term involving $M^\sigma$ equals zero from the Martingale property. The term involving $B^\sigma$ is zero due to Assumption 3.4.

We now claim that the term involving $\tilde{G}_i^\sigma$ goes to zero as well. Since $x_{k,i}^\sigma \to x_i^\sigma$ it holds that, by Lemma 7.2, $(x_{k-[d_i^k]_1,1}^\sigma(t), ..., x_{k-[d_i^k]_j,j}^\sigma(t), ..., x_{k-[d_i^k]_N,N}^\sigma) \to \hat{x}^\sigma(t)$ as well. By the upper semicontinuity of the subgradient, it holds that there exists a $g_i(\hat{x}_i^\sigma(t)) \in \partial_i f(\hat{x}_i^\sigma(t))$ such that

$$g_i((x_{k-[d_i^k]_1,1}^\sigma(t), ..., x_{k-[d_i^k]_j,j}^\sigma(t), ..., x_{k-[d_i^k]_N,N}^\sigma))(t)$$

$$\to g_i(\hat{x}_k^\sigma(t))$$

as $\sigma \to \infty$. Thus each term in the sum converges to $g_i(\hat{x}_{k-j}^\sigma(t))$. Now, given $j$, as $k \to \infty$, the boundedness assumptions and stepsize rules imply that $g_i(\hat{x}_{k-j}^\sigma(t)) \to g_i(\hat{x}_k^\sigma(t))$. On the other hand as $k \to \infty$ and $j \to \infty$, $m^j g_i(\hat{x}_{k-j}^\sigma(t)) \to 0$. Thus $\sum_{j=0}^k m^j g_i(\hat{x}_{k-j}^\sigma(t)) \to \frac{1-m^k}{1-m} g_i(\hat{x}_k^\sigma(t)) \to \frac{1}{1-m} g_i(\hat{x}_k^\sigma(t))$, and the claim has been shown.

Thus the weak sense limit of $\lim_{\sigma\to\infty} W_i^\sigma(\cdot) = W_i(\cdot)$ satisfies

$$\mathbb{E}\left[ h(\tau_i(s_j), \hat{x}(\tau_i(s_j)) (W_i(t+\tau) - W_i(t)) \right]$$

and thus by Kushner & Yin (2003, Theorem 7.4.1) is a martingale and is furthermore a constant with probability one by the Lipschitz continuity of $x$ by Kushner & Yin (2003, Theorem 4.1.1). Thus,

$$W(t) = \hat{x}(t) - \hat{x}(0) - \int_0^t g(\hat{x}(s))ds = 0,$$

where $g(\hat{x}(s)) \in \partial f(\hat{x}(s))$, and the conclusion holds.

## 9 CONVERGENCE WITH PROBABILITY ONE

The previous Theorem showed that under the conditions described for the algorithm, there is a weakly convergent subsequence to an invariant set. We can now use the results in Dupuis & Kushner (1989) to infer from weak convergence, probability one convergence of the sequence of iterates.

For this, we shall use the machinery developed in Dupuis & Kushner (1989), which establishes conditions for which a weakly convergent stochastic approximation algorithm approximating a continuous ODE converges with probability one, under certain conditions. One can study the proof structure to quickly reveal that with minor modifications the results carry through. In particular, when $\dot{b}$ appears in the proof, one can replace it with an element of the differential inclusion, and the limit point is replaced by the invariant set. Assumption 2.1 in Dupuis & Kushner (1989) is now associated

with a set-valued map $S(x, T, \phi)$, and by the noise structure of the assumptions, it can easily be seen that $\bar{L}$ exists for all possible values of $x$, $T$ and $\phi$ in the notation of the paper. One can see that the uniqueness appears once in the beginning of the proof of Theorem 3.1 with the existence of this $T_1$ such that the trajectory lies in a specific ball around the limit point for $t \geq T_1$. This can be replaced by the trajectory lying in this ball around the invariant set, for $T_1$ defined as the supremum of such $\hat{T}_1$ associated with every possible subgradient, i.e., element of the DI. Since the subgradient is a compact set and is upper semicontinuous, this supremum exists. Finally, note that Assumption 3.2 is as Assumption 4.1 in Dupuis & Kushner (1989) and thus similarly implies Theorem 4.1 and Theorem 5.3. This proves that as $\sigma \to \infty$, w.p.1 $x^\sigma(\cdot)$ converges to an invariant set of the differential inclusion.

