# OpenReview forum: "Asynchronous Stochastic Subgradient Methods for General Nonsmooth Nonconvex Optimization"
_ICLR.cc/2020/Conference — Reject_

### Official Review · AnonReviewer3 · 2019-10-21
**Official Blind Review #3**

**Rating:** 3

**Review:**

This paper analyzes convergence of asynchronous methods on general non-smooth and non-convex functions (typically arising from deep leaning). Stochastic sub-gradient asynchronous methods are of particular challenging when coupled with complicated hardware behavior of modern NUMA architecture. To validate the analysis, and study the impact of momentum, variable partitioning, numerical experiments on deep learning training are given.

A major concern to me is that the assumptions made in the analyze may be too restrictive. For example, Assumption 3.1.1 regarding the unbiasedness of the stochastic sub-gradients may not hold for since the Clarke sub-differentiation is not additive. The Clarke sub-differentiation of |x| and -|x| are not included in their average which is zero, therefore the Assumption 3.1.1 is not true for tame functions (as cited in the paper) . If this assumption were not true, all the proof arguments based on Martingale differences may not be follow to prove Theorem 3.1.

There are a few typos which make the paper hard to understand. Is the momentum variable  u_i^kc in Algorithm 1 a central variable, as x_i? It seems to be yes since the update needs a lock. But why there is an kc on its index, which is not the case for x_i?

In terms of numerical results, the system specification is not so clear to me why it includes a diverse asynchronous system settings. Even though Figure 1 shows convergence in terms cross-entropy loss, which is not directly related to validate Theorem 3.1 since it is not clear what is going on with x_t.

The paper would be of great interest if the global shared memory asynchronous model is made more precise in the main body of the paper. At a first glace, it is not clear what is the benefit (or what insights) using this model to analyze these algorithms, compared to simplistic models in literature.

**Experience Assessment:**

I have published one or two papers in this area.

**Review Assessment: Checking Correctness Of Derivations And Theory:**

I assessed the sensibility of the derivations and theory.

**Review Assessment: Checking Correctness Of Experiments:**

I assessed the sensibility of the experiments.

**Review Assessment: Thoroughness In Paper Reading:**

I read the paper at least twice and used my best judgement in assessing the paper.

---

> ### Author Response · Authors · 2019-11-14
> **Minor change indicated and reply**
>
> At first, while we acknowledged the unbiased property does not necessarily hold for Clarke subdifferentiation, we thought it'd be an excessively high standard to prove results without this assumption, given that all of the literature on serial stochastic subgradient methods that we are aware makes this assumption. However, we decided to look into relaxing the condition based on the intuition that for problems of interest, the functions are almost everywhere continuously differentiable, and so a  noisy algorithm will in practice always return a gradient. We formalized this with the introduction of a bias term, and a new Assumption 3.3, and a minor modification of the proof of convergence of the main Theorem in the Appendix.
>
> We fixed some of the typos outstanding in the paper, and in particular clarified the algorithm indices and memory locations.
>
> We decided to test a variety of asynchronous schemes in order to investigate any differentiating properties of them, given that these are the main possibilities available. Block partitioning and lock-free read but lock-free write are both addressed by the convergence theory.
> The objective function shows a leveling out in the figures, which suggests the iterates are converging. By contrast, for nonconvex problems something like $\|x^k-x^*\|$ is known to not be a very good proxy for the measure of convergence.
>
> Finally, the paper adapts a state of the art probabilistic model of asynchronous computation to the stochastic approximation framework. The probabilistic model is given in the Canelli et al. reference, wherein it is explained is considerable detail the major drawbacks of other models and how they are grossly inacurate in capturing the properties of asynchronous parallel computation, and detailing the properties and demonstrating the generality of the new model.

---

> ### Author Response · Authors · 2019-11-15
> **Comment on the numerical results**
>
> Thanks for your comments.
>
> The convergence behavior of the proposed method over a diverse set of asynchronous system settings provides detailed information to a practitioner about the nuances of the implementation with regards to an end-to-end performance.  In particular, we aimed to highlight that when asynchrony was reduced and model size increased, the block-partitioned approach, i.e. PASSM, performed at par with a full gradient updating method WIASSM: the case of ResNet50/CIFAR100 over an asynchronous system with 4 processes using a single GPU per process. Nevertheless, we have now presented another implementation, keeping the system settings and model architecture unchanged, which shows a clear advantage of the proposed method over the compared ones. Please refer to the "Comments on the experiments" above in rebuttal to Review 2.
>
>
> Thank you for indicating a need for a highlight on the system specifications. We have updated the draft clearly marking the specification of the asynchronous shared-memory system: the paragraph "Asynchronous System Specification" in Section 2.

---

### Official Review · AnonReviewer1 · 2019-10-22
**Official Blind Review #1**

**Rating:** 3

**Review:**

This paper proposes a model to study asynchronous stochastic subgradient methods for minimizing a nonsmooth nonconvex function. Studying stochastic subgradient methods in the nonsmooth nonconvex setting is already very challenging. Throwing asynchronous updates into the mix further complicates the analysis. This is overcome by proposing a model for asynchrony which captures the salient features of computation platforms, while being amenable to analysis.

The main drawback of this paper is that the experiments seem to suggest that, although asynchronous methods do a good job optimizing training metrics (loss, accuracy), the models they train do not generalize as well as using synchronous SGD. This demotivates the need for the theory developed in this paper - rather, the theory is ahead of it's time. The paper could be significantly strengthened by reporting an example application where using an asynchronous stochastic subgradient method is advantageous for training deep networks.

The notation and discussion in Sec 3.1 is pretty heavy. Can you explain how this model differs from other models of asynchrony, e.g., as put forward in the well-known book by Bertsekas and Tsitsiklis on Parallel and Distributed Computation?

The main convergence result (Theorem 1) holds for a continuous-time process, following the typical analysis of stochastic approximation algorithms. However, practitioners rarely use a diminishing step size, especially for training deep networks. Is it possible to quantify the effect of using a constant step size? (E.g., in stochastic approximation analyses for smooth functions, one typically gets a "law of large numbers" which ensures convergence of the average, and a "central limit theorem" which bounds the distance of any realization of the process to its mean. Is there an analogous CLT for this setting?)

In the experiments, the SGD with DataParallel is effectively using a mini-batch size that is 3-5x larger than that of a single worker. In this case one would expect to be able to use a larger learning rate too (since there is less noise). Evidence for this is provided in Goyal et al., "ImageNet in 1 hour". Did you try tuning the learning rate separately for each method?

How were the parameters of the deep CNNs used in the experiments divided into blocks? This choice should affect the average time to evaluate a block gradient, right? Computing the gradient of parameters in the lowest layer requires doing a full forward and backward, while computing the gradient of higher layers should be faster since it doesn't require a full backward pass. Did you measure the average time to compute a block gradient, and could you report it in the paper? Did you divide into blocks based on layer? Are all parameters within the same block part of the same filter at a given layer?

The last paragraph of Sec 4 talks about the necessity of momentum, but it isn't clear what evidence this claim is based on.




**Experience Assessment:**

I have published one or two papers in this area.

**Review Assessment: Checking Correctness Of Derivations And Theory:**

I assessed the sensibility of the derivations and theory.

**Review Assessment: Checking Correctness Of Experiments:**

I carefully checked the experiments.

**Review Assessment: Thoroughness In Paper Reading:**

I read the paper thoroughly.

---

> ### Author Response · Authors · 2019-11-15
> **Comment on the experiments**
>
> Thank you for your time in reading the paper, and we appreciate that you recognize the importance and challenge of this work.
>
> Thanks indeed for your specific valuable comments on the experiments which motivated us to have a deeper look at the loopholes in the implementation strategy which remained a bit short of being able to experimentally highlight the significance of a novel theoretical result.
>
> We updated our implementation and experiments, which adopts a different strategy to overcome the challenges of implementation of the asynchronous methods. In our updated implementation, concurrent processes get equal opportunity to benefit from the high-performance data processing of a multi-GPU system. Now, on comparing the end-to-end convergence speedup with respect to wall-clock-time, the proposed PASSM method outperforms the compared existing methods.
>
> -- Specification of the updates:
>   The updated implementation of the asynchronous methods works as the following: each of the concurrent processes performs both the forward and the backward passes in a data-parallel way utilizing all the available GPUs, thereby taking advantage of a multi-GPU system and the parallelization offered therein. It effectively simulates a shared-memory system over a "distributed" memory setting.
>
> -- Mini-batch-size and other hyperparameters:
>   Data-parallel SGD could process a larger mini-batch size utilizing a larger amount of parallel resources at disposal resulting in a reduction of variance. However, this will be a bit unfair to the asynchronous methods which would remain devoid of this statistical advantage while employing concurrent processes. For fairness’ sake, we used identical hyper-parameters for each of the methods. Examining closely, SGD will actually have a smaller processing time per mini-batch as it would use more CUDA threads for an identically sized mini-batch compared to its asynchronous counterparts. Please note that Goyal et al., "ImageNet in 1 hour" explore the advantage of larger mini-batch processing that could significantly save on the communication cost. In our case, it is a shared-memory setting which does not consider communication cost. In particular, in the updated implementation, we did not allow the concurrent processes to perform their iterations in isolation on separate GPUs, for they exploit data-parallelization as much as SGD. Counting the total number of mini-batches processed, now all the processes are almost at an equal footing on the count of cross-GPU communication costs.
>
> -- Load-balancing for back-propagation:
>  It is imperative that there will be a difference in terms of time taken for a forward and a backward pass by different processes depending on the proximity of their respective assigned blocks to the output layer. However, note that the multi-GPU setting would provide the concurrent processes with the utilization of the available parallel resources depending on the number of tensors that it needs to process, effectively resulting in almost a load-balancing. Nevertheless, we assigned blocks containing $s_i \ge \lceil{L/p}\rceil$ parameter components,  where $L$ is the model size and $p$ is the number of processes, to the processes $P_i$ computing stochastic sub-gradients corresponding to layers closer to output. Whereas, the process that computes sub-gradient of the layer-block closest to the input is assigned a block containing less than $\lfloor{L/p}\rfloor$ parameter components. The assignments $s_i$ aim to balance computation load, however, it varies across layers depending on the size of the assigned leaves in terms of parameter component. However, we observed an insignificant difference in average processing time for the same number of epochs irrespective of switching off the requires_grad flag inferring that a blocked architecture such as ResNet does not allow much scope of computation-cost saving on this count. We did not emphasize on assigning the weight and bias tensors of layers to the same process mainly to avoid unnecessary "load im-balance" when they all are using data-parallel resources. In any case, $p$ is much smaller than $L$ so there would not be more than just a few cases of tensors of a layer landing on different processes.
>
> -- On momentum:
>  Finally, now we have a specific set of benchmarks that highlight the significance of momentum. In particular, it shows that the asynchronous methods take better advantage of momentum correction which was also observed by Mitliagkas et al, “Asynchrony begets momentum” 2017.
>
> We have significantly updated the draft clearly highlighting the above-mentioned points.

---

> ### Author Response · Authors · 2019-11-15
> **Response and edit**
>
> Bertsekas and Tsitsiklis were the first to consider asynchrony but concentrated on strictly the smooth and convex case. They consider several models of asynchrony, one with strong assumptions on the scheduling and with stronger results, and one very generic set of conditions, but with convergence on a set of problems satisfying a condition even stronger than strong convexity (diagonally dominant Hessian eigenvalues). There were quite a few works modeling asynchronous opitimization in recent years starting with the well known Hogwild! paper. However, they assumed complete uniformity in the computational capabilities of different cores, and one of the achievements of the cited work of Canelli et al. is generalizing the class of asynchronous methods to be potentially analyzed in relaxing these conditions, and this probabilistic model is what we build off of, as explained in the appendix.
>
> With a constant step size, stochastic approximation under more challenging scenarios typically will have a weak convergence result, where the entire path of the sequence of iterates approaches the continuous path that is the differential inclusion solution, as the constant stepsize approaches zero. We feel that such a result is less intuitively comprehensible / meaningful by the community of potential readers of this work, compared to almost sure convergence to a stationary point.
> More sophisticated results for constant stepsizes, such as non-asymptotic results, convergence in expectation and approximation by a stochastic diffusion process with a central limit argument, are not available in the case of nonsmooth optimization even in the serial case, without any parallelism or asynchrony.

---

### Official Review · AnonReviewer2 · 2019-10-25
**Official Blind Review #2**

**Rating:** 3

**Review:**

This paper provides a convergence analysis for asynchronous optimisation in the case where non-smoothness imposes the use of sub gradients and non-convexity (combined with asynchrony) introduces more challenging staleness (delay) issues than in the convex case.

strength:
the paper tackles one of the hardest settings to be analysed in distributed optimisation, and packages it in a readable continuous-time framework.

weakness:
as far as I went into the details, I couldn't understand how the authors tackle one the biggest problem in non-convex + asynchrony: coherence of gradients.

Specifically, my sole question to the authors is how does their analysis take into account (sub)gradients that are delayed *and* not in the same half-space as the current non-biased estimator of the gradient ? These (sub)-gradients will act in an almost malicious manner. (as usual in distributed computing, where asynchrony can be modelled by a malicious scheduler).
Maybe my reading made me miss where this is handled, if a precise pointer is given I will upgrade my score.

**Experience Assessment:**

I have published in this field for several years.

**Review Assessment: Checking Correctness Of Derivations And Theory:**

I assessed the sensibility of the derivations and theory.

**Review Assessment: Checking Correctness Of Experiments:**

I did not assess the experiments.

**Review Assessment: Thoroughness In Paper Reading:**

I read the paper at least twice and used my best judgement in assessing the paper.

---

> ### Author Response · Authors · 2019-11-14
> **Comment on gradient incoherence**
>
> Thank you for taking the time to read the paper, and we appreciate that you recognize the importance and challenge of this work.
>
> The assumptions on both the form of the algorithmic updates, as well as the probabilistic model of asynchronous scheduling should be general enough such that it encompasses even horribly adversarial subgradient estimates. The convergence theory does show that indeed although the gradient estimates are based on delayed information, and furthermore in the read-lock-free setting, they can be estimates at points which never ever existed in memory altogether (but their components did, at different times/iterations), the basic stochastic subgradient algorithm does converge.
>
> So in answer to the main question, the assumptions of the theory is general enough to include the worst possible scenarios in delayed inaccurate subgradient estimates. Of course, we'd like to also offer some guidance in finding the intuition in the theory so as to help explain how it is taken care of, in reference to the convergence theory on page 13.
>
> An essential element here is the gradual stepsize decrease together with a maximum possible delay. Asymptotically, due to the decreasing stepsize, the iterates get closer and closer together, so the delayed gradients become an ever better approximation to the current ones. Eventually they are accurate enough that the error associated with the delays is outweighed by the progress in minimizing the function associated with the subgradient.

---

### Author Response · Authors · 2019-11-15
**Revision Summary**

We again thank the reviewers for their insightful comments. We review the significant changes in this revision:

- We have overhauled our implementation and are now able to provide practical end-to-end convergence speedup for PASSM versus SGD and other asynchronous methods!

- We therefore present the first distributed algorithm which provides both theoretical guarantees, as well as practical speedup in the challenging asynchronous non-convex non-smooth case

- We have addressed Reviewer 2's question regarding "adversarial" delayed sub-gradients

- We resolved the issue regarding Clarke sub-differentiation via the introduction of a bias term (see Assumption 3.3), and a minor modification of the proof of convergence of the main Theorem.

- We have clarified the system specification and the relation with previous work

- We have undertaken a major revision of the paper to address all the outstanding reviewer comments regarding clarity and significance

---

### Decision · Program_Chairs · 2019-12-19

**Decision:**

Reject

**Comment:**

This paper considers an interesting theoretical question. However, it would add to the strength of the paper if it was able to meaningfully connect the considered model as well as derived methodology to the challenges and performance that arise in practice.